# European polygenic risk score for prediction of breast cancer shows similar performance in Asian women

Weang-Kee Ho [iD] et al.[#]

Polygenic risk scores (PRS) have been shown to predict breast cancer risk in European women, but their utility in Asian women is unclear. Here we evaluate the best performing PRSs for European-ancestry women using data from 17,262 breast cancer cases and 17,695 controls of Asian ancestry from 13 case-control studies, and 10,255 Chinese women from a prospective cohort (413 incident breast cancers). Compared to women in the middle quintile of the risk distribution, women in the highest 1% of PRS distribution have a ~2.7-fold risk and women in the lowest 1% of PRS distribution has ~0.4-fold risk of developing breast cancer. There is no evidence of heterogeneity in PRS performance in Chinese, Malay and Indian women. A PRS developed for European-ancestry women is also predictive of breast cancer risk in Asian women and can help in developing risk-stratified screening programmes in Asia.

[#]A list of authors and their affiliations appears at the end of the paper.

In the majority of high-income Western countries, breast cancer screening is systematic and population-based, and this has contributed to an improvement in survival[1]. By contrast, screening in the majority of Asian countries is opportunistic and suffers from poor uptake, contributing to delayed detection and poor survival[2]. In addition, there are concerns about the appropriate starting age of screening, as women are recommended to start screening at age 50 in many Asian countries, even though the peak breast cancer incidence in Asian populations is between 40 and 50 years of age[3]. Taken together with the rapidly increasing incidence of breast cancer in Asia[4], there is thus an urgent need to develop an appropriate screening strategy for Asian women.

Provision of genetic counselling and genetic testing for rare variants in breast cancer predisposition genes such as *BRCA1* and *BRCA2* can lead to better management of risk, but these only explain a small fraction of breast cancer cases in the general population[5]. Risk profiles based on a combination of low penetrance but common breast cancer susceptibility single nucleotide polymorphisms (SNPs), summarised as polygenic risk scores (PRS), have been shown to be an important predictor of disease risk[6–8]. A 313-SNP PRS developed in European populations has improved predictive power compared to earlier PRS based on fewer SNPs;[6,7] this PRS demonstrated similar associations with disease risk in eleven independent prospective studies. Studies in European populations have demonstrated that PRS substantially improve discrimination, in comparison to risk prediction models based on classical risk factors alone[8,9]. In particular, using the recent extension of the BOADICEA model, it has been demonstrated that the 313-SNP PRS provides greater level of risk stratification in the population than epidemiological risk factors alone, and that the greatest level of risk stratification is achieved when both the PRS and epidemiological risk factors are considered jointly[10]. Screening trials[11–13] in women of predominantly European descent are ongoing to evaluate personalised breast cancer screening programme based on a woman's individual risk of disease, as a means of improving screening efficiency[14].

Although there have been several efforts to create an Asian-specific PRS, these have been limited by the smaller sample size of Asian genetic studies[15–19]. Only ~20% of the existing breast cancer genome-wide association study (GWAS) data are from women of Asian ancestry[20,21]. This limits the precision in the relative risk estimates for individual variants, which is critical for development of predictive PRS. Furthermore, Asian populations are ethnically and genetically diverse[22], and genetic associations with breast cancer risk may vary by ancestry. Here, we evaluate the predictive ability of the 313-SNP PRS developed for European women for predicting breast cancer risk in Asian women, using data from 17,262 cases and 17,695 control women of Asian ancestry, from 10 studies based in Asian countries and three studies from North America, participating in the Breast Cancer Association Consortium (BCAC); and 10,255 Chinese women from a prospective cohort. We also evaluate the heterogeneity in the associations with breast cancer risk by ethnicity. We show that European ancestry-based PRS is predictive of breast cancer risk in Asian women.

## Results

### SNPs included in PRS analyses.
To ensure accurate determination of PRS in the ethnic-specific analyses, 26 of the 313 SNPs with imputation accuracy scores <0.9, based in the Malaysian Breast Cancer Genetic Study (MyBrCa) and Singapore Breast Cancer Cohort (SGBCC) of 6900 cases and 7606 controls, combined, were excluded. Hence, the PRS was constructed using 287

SNPs for all BCAC studies (Supplementary Table 1). For the Singapore Chinese Health Study (SCHS), 229 of the 287 SNPs that were polymorphic and could be imputed in this dataset were used for PRS derivation. To compare the PRS performance with that in women of European ancestry, we recalculated the PRS using these sets of SNPs in the validation and prospective cohorts of European women described in Mavaddat et al.[7].

For association analyses between PRS and overall breast cancer, the PRS was calculated using overall breast cancer weights while for association analyses between PRS and subtype-specific breast cancer, subtype-specific PRSs were constructed using the same set of SNPs but weights from the hybrid method described by Mavaddat et al.[7] [see section "Methods"]. The list of SNPs and corresponding weights used to construct the 287-SNP PRS and 229-SNP PRS are provided in Supplementary Data 1.

### PRS and breast cancer risk in Asian women living in Asia.
Data on 15,755 invasive cases and 16,483 control women from 10 Asian studies in BCAC were included (Supplementary Table 1). The mean of the 287-SNP PRS was markedly higher in Asian women compared to European women for overall breast cancer PRS ($PRS_{OVERALL}$), ER-positive PRS ($PRS_{ER+}$) and ER-negative PRS ($PRS_{ER-}$), while the standard deviations (SDs) were slightly lower in Asian controls [versus European controls] for all three PRSs (0.556 [0.597], 0.592 [0.638] and 0.533 [0.567], respectively, Table 1). The remaining analyses for 287-SNP PRSs in this manuscript are presented in terms of the PRS standardised to the SD in the European controls.

Table 2 shows the estimated odds ratio (OR) per unit increase of standardised PRSs for overall and subtype-specific breast cancer. For overall breast cancer, the estimated OR per SD was 1.52 (95% confidence interval (CI): 1.49–1.56). For subtype-specific disease, the estimated OR per SD of the subtype-specific PRS was 1.62 (95% CI: 1.57–1.67) for ER-positive and 1.41 (95% CI: 1.36–1.46) for ER-negative disease. There was no evidence of heterogeneity among studies genotyped with either the iCOGS or OncoArray ($p$ values for heterogeneity >0.05 [chi-squared test], Fig. 1). For overall breast cancer and ER-positive disease, the ORs per SD of the PRS were slightly higher in OncoArray genotyped studies compared to the studies genotyped with iCOGS. However, the confidence intervals for the array-specific estimates overlapped (Fig. 1). There was no evidence that the effect of PRS was modified by age ($p$ value of interaction < 0.05 [Student's $t$ test]; Supplementary Table 2). When analyses were stratified by 10-year age groups, the ORs per SD of the PRS by age group were similar (Supplementary Table 3).

The association between the PRSs and breast cancer risk by PRS percentile are shown in Fig. 2 and Supplementary Table 4. Compared to women in the middle quintile (40–60%), the observed OR of developing overall breast cancer for women in the highest and lowest 1% of the PRS distribution was 2.72 (95% CI: 2.24–3.29) and 0.38 (95% CI: 0.27–0.52), respectively. Women in the highest and lowest 1% of the ER-specific PRSs had 2.84 (95% CI: 2.30–3.49)- and 0.25 (95% CI: 0.16–0.39)-fold risk, respectively, for ER-positive disease, and 2.29 (95% CI: 1.77–2.97)- and 0.57 (95% CI: 0.36–0.90)-fold risk, respectively, for ER-negative disease. The observed ORs by PRS percentile did not differ from those predicted under a theoretical polygenic model in which the log OR depends-linearly on the PRS: all predicted ORs fall within the confidence intervals of the observed ORs (Fig. 2; Supplementary Table 4).

Table 3 shows the association between family history of breast cancer and overall/ER-specific breast cancer risk, adjusted and unadjusted for standardised overall/ER-specific PRSs. Family history information was not available for all cases in Seoul Breast

**Table 1 Mean and standard deviation of 287-SNP and 229-SNP polygenic risk scores.**

| | Cases, N | Controls, N | Mean PRS (SD) 287-SNP PRS Cases | Mean PRS (SD) 287-SNP PRS Control | Mean PRS (SD) 229-SNP PRS Cases | Mean PRS (SD) 229-SNP PRS Control |
|---|---|---|---|---|---|---|
| **Asian studies in BCAC** | | | | | | |
| PRS$_{OVERALL}$ | 15,755 | 16,483 | 0.91 (0.554) | 0.69 (0.556) | 1.09 (0.537) | 0.88 (0.539) |
| PRS$_{ER+}$ | 10,477 | 16,483 | 0.89 (0.580) | 0.62 (0.592) | | |
| PRS$_{ER-}$ | 4764 | 16,483 | 1.26 (0.543) | 1.07 (0.533) | | |
| **Asian within American studies in BCAC** | | | | | | |
| PRS$_{OVERALL}$ | 1507 | 1212 | 0.91 (0.560) | 0.75 (0.546) | 1.09 (0.545) | 0.93 (0.529) |
| PRS$_{ER+}$ | 1022 | 1212 | 0.89 (0.598) | 0.69 (0.583) | | |
| PRS$_{ER-}$ | 280 | 1212 | 1.27 (0.513) | 1.12 (0.512) | | |
| **Prospective cohort** | | | | | | |
| PRS$_{OVERALL}$ | 413 | 9842 | | | 1.06 (0.539) | 0.85 (0.523) |
| **European studies** | | | | | | |
| PRS$_{OVERALL}$ | 5129 | 5285 | 0.44 (0.608) | 0.12 (0.597) | 0.72 (0.608) | 0.435 (0.556) |
| PRS$_{ER+}$ | 4233 | 5285 | 0.43 (0.651) | 0.05 (0.638) | | |
| PRS$_{ER-}$ | 926 | 5285 | 0.78 (0.560) | 0.54 (0.567) | | |

The overall breast cancer (BC) PRS (PRS$_{overall}$) and oestrogen-receptor (ER)-positive PRS (PRS$_{ER+}$) and ER-negative PRS (PRSER−) were derived as describe in the "Method" section. ER-subtype is not available for the prospective cohort. Mean and SD of PRS in European studies were calculated using the data on the validation set as described in Mavaddat et. al.[7] but samples with missing ages information were removed. *BCAC* Breast Cancer Association consortium, *SD* standard deviation, *PRS* polygenic risk scores.

**Table 2 Association between standardised polygenic risk scores and breast cancer risk.**

| | Cases, N | Controls, N | 287-SNP PRS OR per SD$^a$ (95% CI) | 287-SNP PRS AUC | 229-SNP PRS OR per SD$^a$ (95% CI) | 229-SNP PRS AUC |
|---|---|---|---|---|---|---|
| **Asian studies in BCAC** | | | | | | |
| Overall BC | 15,755 | 16,483 | 1.52 (1.49–1.56) | 0.613 | 1.49 (1.45–1.52) | 0.611 |
| ER-positive | 10,477 | 16,483 | 1.62 (1.57–1.67) | 0.627 | | |
| ER-negative | 4,764 | 16,483 | 1.41 (1.36–1.46) | 0.594 | | |
| **Asians within North American Studies in BCAC** | | | | | | |
| Overall BC | 1507 | 1212 | 1.36 (1.25–1.49) | 0.577 | 1.33 (1.22–1.45) | 0.579 |
| ER-positive | 1022 | 1212 | 1.38 (1.25–1.53) | 0.586 | | |
| ER-negative | 280 | 1212 | 1.49 (1.26–1.76) | 0.587 | | |
| **Prospective cohort** | | | | | | |
| Overall BC | 413 | 9842 | | | 1.49 (1.33–1.67) | 0.61 |
| **European studies** | | | | | | |
| Overall BC | 11,225 | 17,788 | 1.61 (1.57–1.66) | 0.630 | 1.59 (1.55–1.64) | 0.627 |
| ER-positive | 7809 | 17,788 | 1.68 (1.64–1.73) | 0.642 | | |
| ER-negative | 1234 | 17,788 | 1.44 (1.36–1.53) | 0.600 | | |

$^a$Adjusted for first ten principal components and study, and standardised to SDs of PRSs in European controls as shown in Table 1. For prospective cohort, model was adjusted for first seven principal components. Only 229 of the 287 SNPs that were polymorphic and could be imputed were available for the prospective cohort. To enable comparison between case-control studies and prospective cohort, we included the results of 229-SNP PRS for all studies. For studies in Breast Cancer Association Consortium (BCAC), AUCs were adjusted by study. The OR per SD and AUC for European studies were estimated using the same data on the prospective cohorts as described in Mavaddat et al.[7]. *PRS* polygenic risk scores.

Cancer Study (genotyped on OncoArray) or control women in the second batch of Singapore Breast Cancer Cohort, hence both studies were excluded from these analyses. The percentage attenuation in the log ORs for family history after adjusting for PRSs was 10.0% for overall breast cancer (unadjusted family history OR = 1.35, adjusted OR = 1.31), 7.3% for ER-positive breast cancer (unadjusted OR = 1.36, adjusted OR = 1.33) and 13.2% for ER-negative breast cancer (unadjusted OR = 1.2, adjusted OR = 1.18). There was no evidence of interaction between the PRSs and family history (p values ≥ 0.05 [Student's t test], Supplementary Table 2). Including family history in the model, in addition to the PRS, increased the AUC only slightly (0.616 vs. 0.613 for PRS alone; Table 2).

**PRS and breast cancer risk in Chinese, Malays and Indians.** Analyses by ethnic subgroup were limited to 6900 invasive cases

and 7506 controls participating in MyBrCa and SGBCC studies (Supplementary Table 1). Malaysia and Singapore are ethnically diverse, with the majority of individuals identifying as Chinese, Malay or Indian. Principal component analysis showed that these ethnic groups can be distinguished based on genetic data; however, the distribution of the first two principal components for each ethnic group was similar between the two countries (Supplementary Fig. 1). Hence for the purposes of this analysis, women belonging to the same ethnic group from the two countries were analysed together.

Table 4 summarises the characteristics of the study participants by self-reported ethnicity. The majority of the participants were Chinese (72%), while 17% and 11% were Malay and Indian, respectively. The mean PRS was markedly higher in Chinese and Malay women compared to European women, with the mean being highest for Chinese women. The mean for Indian women was intermediate between those for Chinese and Malay women

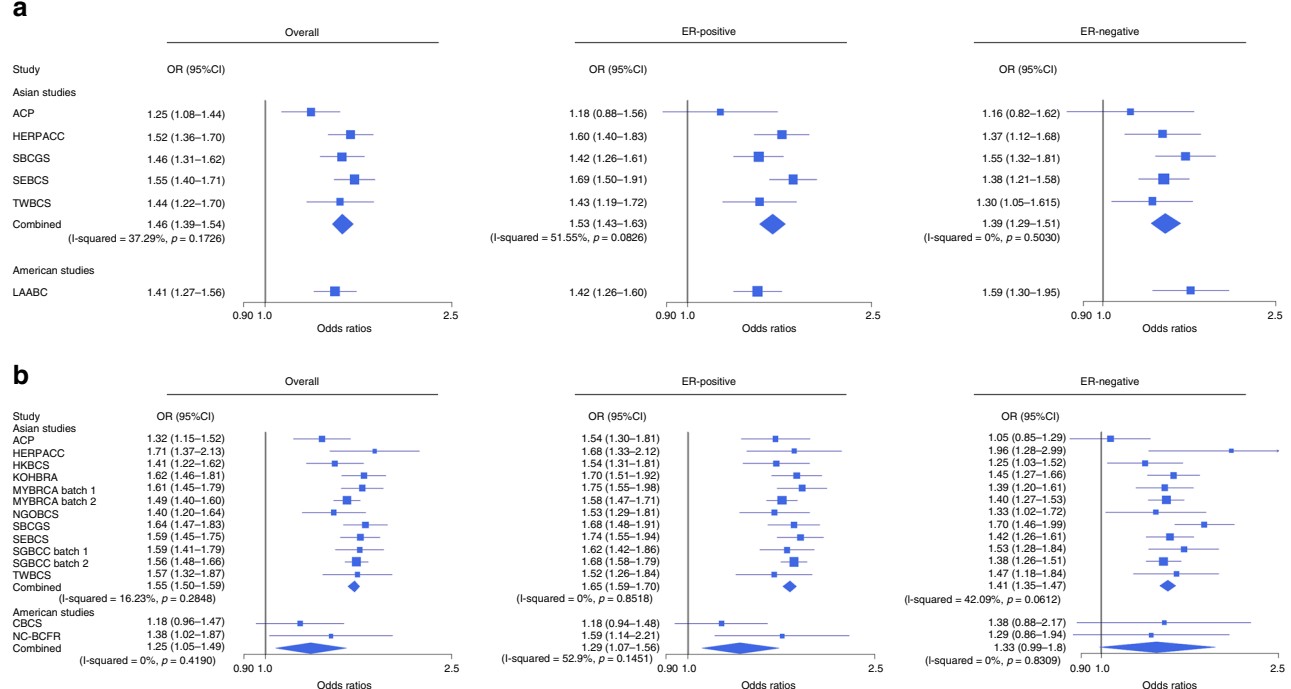

**Fig. 1 Association between standardised 287-SNP polygenic risk scores and breast cancer risk.** Panel **a** shows the results for iCogs array by study and panel **b** shows the results for Oncoarray. The squares represent the odds ratios (ORs) and the horizontal lines represent the corresponding 95% confidence intervals. Overall estimates within genotyping array were obtained by combining the estimates across studies using fixed-effect meta-analysis, represented by the diamond shape. I-squared and p value (two-sided) for heterogeneity were obtained by fitting a random-effects model and using generalised Q-statistic estimator (the *rma()* command in R). The sample size of individual studies are listed in Supplementary Table 1. The ORs and corresponding 95% confidence intervals are provided as a Source Data file.

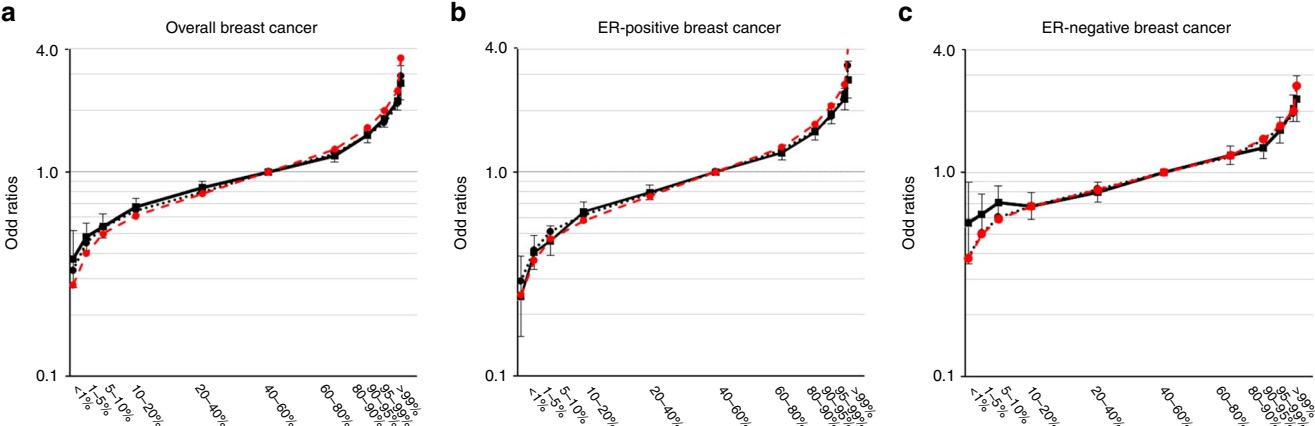

**Fig. 2 Association between percentiles of 287-SNP polygenic risk scores (PRS) and breast cancer risk in combined Asian studies.** The results for overall breast cancer, oestrogen-receptor (ER)-positive breast cancer and ER-negative breast cancer are shown in Fig. 2a–c, respectively. The squares/dots represent the odds ratios (ORs) and the vertical lines represent the corresponding 95% confidence intervals, with middle quintile (40–60th) as the reference category. Solid lines represent the observed ORs, black dashed lines represent the predicted ORs of PRSs under a multiplicative polygenic model in the Asian population and the red dashed line represent the predicted OR in the European population. The analysis was conducted using 15,755 cases and 16,438 controls. Of 15,755 cases, 9989 were ER-positive breast cancer while 4611 were ER-negative breast cancer. Source data are provided in Supplementary Table 5.

and those for European women (Tables 1 and 4). The PRS SDs of Malay and Indian controls were similar to that of women of European ancestry, while Chinese's SDs were slightly lower.

The breast cancer OR per SD of the 287-SNP PRSs and the discriminatory accuracy, measured by area under the receiver operating characteristic curve (AUC), was similar across the three ethnic groups (heterogeneity p values > 0.05 [chi-squared test]; AUCs for overall breast cancer were

0.60–0.62, for ER-positive disease were 0.62–0.63 and for ER-negative disease were 0.57–0.60; Fig. 3). OR estimates by percentiles for overall breast cancer risk, compared to the middle quintile are shown in Fig. 4 and Supplementary Table 5. The OR estimates were similar across ethnicities, except that for the highest 10% of the PRS distribution, where Chinese had a higher OR (2.19, 95% CI: 1.91–2.52) compared to Malays (1.79 95% CI: 1.35–2.37) and Indians (1.57, 95% CI: 1.09–2.26).

However, the confidence intervals of the ethnicity-specific estimates overlapped (Fig. 4).

**PRS and breast cancer risk in Asian Americans**. The 287-SNP PRS was also evaluated using data from 2719 women of Asian ancestry recruited into three studies from North America (Supplementary Table 1). The means for all PRS were very similar to those in the Asian studies, and markedly higher than those in Europeans. The SDs in controls for all PRSs were similar to those in the Asian studies and somewhat lower than the observed SDs in European controls (Table 1).

Compared to the breast cancer OR per SD in the Asian studies from Asia, the OR per SD of the 287-SNP PRS in the North American studies was smaller ($p < 0.05$) for overall breast cancer (1.36, 95% CI: 1.25–1.49) and ER-positive breast cancer (1.38, 95% CI: 1.25–1.53), but higher ($p < 0.05$) for ER-negative breast cancer (1.49, 95% CI: 1.26–1.76, Table 2). Of the three studies included in these analyses, only the Los Angeles County Asian–American Breast Cancer (LAABC) case–control study showed a significant association with breast cancer risk for all three PRSs while the Canadian Breast Cancer (CBC) study showed non-significant association across all PRSs (Fig. 1). However, the heterogeneity in the estimates among studies was not significant.

**Prospective evaluation for PRS**. We further evaluated the PRS in the prospective Singapore Chinese Health Study (SCHS), using data on 10,255 women, of whom 413 had developed breast cancer (Supplementary Table 1). The mean and SD of the 229-SNP PRS in the prospective study were similar to the mean and SD of 229-SNP PRS in the BCAC Asian studies (Table 1). The estimated

hazard ratio (HR) for overall breast cancer, per European-SD of the 229-SNP PRS, was 1.49 (95% CI: 1.33–1.67) and the AUC was 0.610 (Table 2). The estimates were similar to those for the 229-SNP PRS in Asian studies (Asian studies from Asia: 1.49 (1.45–1.52); from North American studies: 1.33 (1.22–1.45)) but slightly lower than those in the European studies (1.59 (1.55–1.64)).

**Absolute risk of developing breast cancer by PRS percentiles**. Absolute lifetime and 10-year breast cancer risks by 287 SNP PRS percentile were derived by combining the estimated overall breast cancer ORs from BCAC Asian studies (Supplementary Table 4) and the breast cancer incidence and mortality rates for Chinese, Malay and Indian women in Singapore[23,24] (Table 5; Supplementary Fig. 2). The risks of developing breast cancer by age 80 for women in the lowest and highest 1% of the PRS distribution were ~2% and ~13–16%, respectively, depending on ethnicity. For women between the 90 and 99th percentiles of the risk distribution, the lifetime risks vary from 9 to 13%. Assuming that a 10-year absolute risk threshold of 2.3% (approximately the 10-year risk from age 50 in women of European descent[25]) is used to define women at sufficient risk to justify screening, Chinese and Malay women in the highest 1% of the PRS distribution would reach this threshold by age 35, while Indian women in the highest 1% would reach the threshold at age 39 years.

We also determined the proportion of women in the general population who would have 10-year absolute risk above the risk threshold (2.3%) at some point in their life. The maximum 10-year absolute risk for Chinese women in the highest 25%, Malay women in the highest 16% and Indian women in the highest 17% of the PRS distribution were greater than 2.3%. Offering screening to these women would capture ~40%, ~27% and ~28% of all breast cancer cases in the Chinese, Malay and Indian populations, respectively (Supplementary Fig. 3).

**Comparison with other PRSs**. We compared the predictive performance of the 287-SNP PRS for overall breast cancer with five PRSs[15,17,19,26,27], which were previously developed or evaluated using data from Asian populations. Of these 5 PRSs, one was developed using iCogs genotyped studies in BCAC and 744 samples from MyBrCa study[15]. To avoid the potential of overfitting and to enable direct comparison between PRSs, we limited the analyses to OncoArray genotyped studies only (excluding 744 samples from MyBrCa study). We also recalculate the 287-SNP PRS using the same samples. The list of SNPs and corresponding weights as reported in the literature are given in Supplementary Table 6. The ORs per one SD of the 5 Asian PRSs

---

**Table 3 First-degree family history of breast cancer and breast cancer risk in Asian studies.**

|  | Unadjusted for PRS, OR* (95% CI) | Adjusted for PRS, OR (95% CI) | Attenuation (%)[a] |
|---|---|---|---|
| Overall BC | 1.35 (1.22–1.48) | 1.31 (1.18–1.45) | 10.0 |
| ER-positive | 1.36 (1.22–1.53) | 1.33 (1.19–1.49) | 7.3 |
| ER-negative | 1.21 (1.05–1.39) | 1.18 (1.03–1.37) | 13.2 |

PRS was computed based on 287 SNPs. *Odds ratio for developing breast cancer for women with a family history of breast cancer in a first-degree relative compared with women without a family history, adjusted for study and 10 principal components. All the case-control studies listed in Supplementary Table 1 were included in this analysis, except for SEBCS and SGBCC Batch 2. Family history information was not available for all cases in SEBCS (genotyped on Oncoarray) and all controls for SGBCC Batch 2 hence excluded from the analyses.
[a]Percent attenuation on log scale.

---

**Table 4 Characteristics of women in Malaysian Breast Cancer Genetic Study and Singapore Breast Cancer Cohort.**

|  | Chinese Cases | Chinese Controls | Malay Cases | Malay Controls | Indian Cases | Indian Controls |
|---|---|---|---|---|---|---|
| *N* |  |  |  |  |  |  |
| Overall BC | 5236 | 5156 | 1084 | 1332 | 580 | 1018 |
| ER-positive | 3627 | 5156 | 715 | 1332 | 374 | 1018 |
| ER-negative | 1365 | 5156 | 336 | 1332 | 184 | 1018 |
| Age, years, mean (SD) | 53.1 (10.93) | 51.8 (9.73) | 49.7 (10.18) | 51.1 (8.73) | 53.2 (10.53) | 54.1 (8.75) |
| Polygenic risk score, mean (SD) |  |  |  |  |  |  |
| PRS$_{overall}$ | 0.91 (0.54) | 0.69 (0.54) | 0.85 (0.58) | 0.63 (0.58) | 0.56 (0.57) | 0.33 (0.61) |
| PRS$_{ER+}$ | 0.89 (0.57) | 0.63 (0.58) | 0.84 (0.59) | 0.57 (0.62) | 0.55 (0.57) | 0.26 (0.65) |
| PRS$_{ER-}$ | 1.24 (0.53) | 1.07 (0.52) | 1.11 (0.55) | 0.98 (0.55) | 0.92 (0.57) | 0.75 (0.55) |

N is the number of samples for cases and control in each subtype and ethnicity. Age is age of consent for controls and age of cancer diagnosis for cases. Mean and SD of age of cancer diagnosis were calculated using overall breast cancer (BC) cases. PRS was computed based on 287 SNPs. Mean and SD of PRS$_{overall}$, PRS$_{ER+}$ and PRS$_{ER-}$ in cases were calculated using all cases, ER-positive cases and ER-negative cases, respectively. Self-declared ethnicity was used.

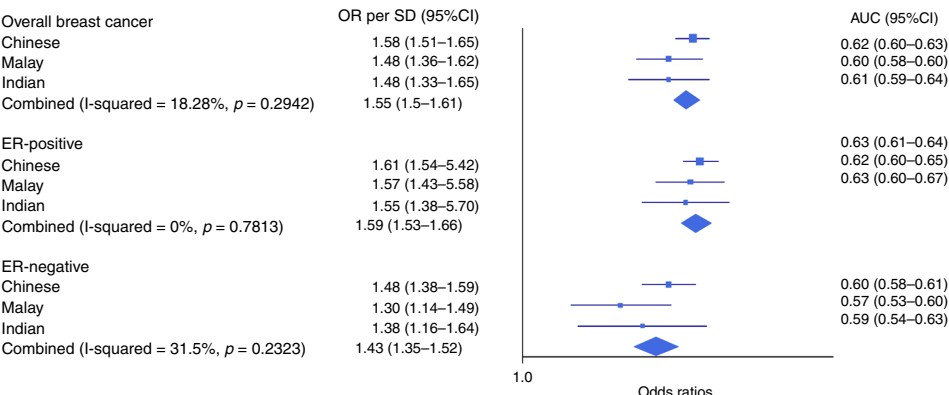

**Fig. 3 Association between standardised PRSs and breast cancer risk in Chinese, Malay and Indian women from Malaysia and Singapore.** Odds ratios (ORs) and AUCs were generated using data from Malaysia Breast Cancer Genetics (MyBrCa) and Singapore Breast Cancer Cohort (SGBCC) studies, stratified by ethnicity. The squares represent the odds ratios (ORs), the horizontal lines represent the corresponding 95% confidence intervals and the diamond shapes represent the overall estimates. I-squared and p value (two-sided) for heterogeneity were obtained by fitting a random-effects model and using generalised Q-statistic estimator (the *rma()* command in R). The number of cases and controls for each ethnicity by breast cancer subtypes are tabulated in Table 4. The sample size, ORs and corresponding 95% confidence intervals are also provided in the Source Data file.

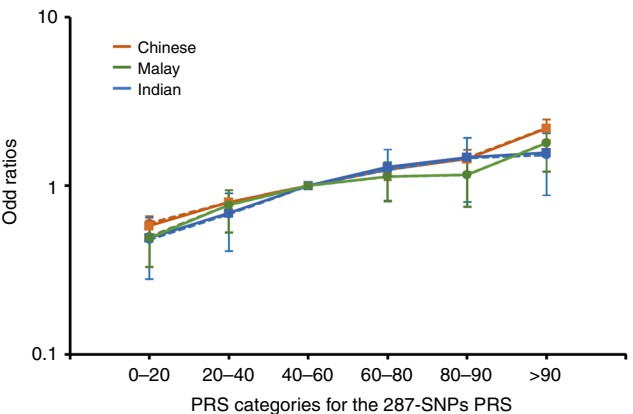

**Fig. 4 Association between percentiles of 287-SNP polygenic risk scores and overall breast cancer risk in Chinese, Malay and Indian women from Malaysia and Singapore.** Results were generated using 5236/5516 Chinese cases/controls, 1084/1332 Malay cases/controls and 580/1018 Indian cases/controls from Malaysia Breast Cancer Genetics (MyBrCa) and Singapore Breast Cancer Cohort (SGBCC) studies, stratified by ethnicity. The squares represent the odds ratios (ORs) and the vertical lines represent the corresponding 95% confidence intervals, with middle quintile (40–60th) as the reference category. Solid lines represent the observed ORs and dashed lines represent the predicted ORs of PRS under a multiplicative polygenic model. Source data are provided in Supplementary Table 6.

were between 1.10 and 1.41 and corresponding AUCs were between 0.533 and 0.586, substantially lower than that for the European-ancestry based 287-SNP PRS (Table 6).

## Discussion

To date, the utility of incorporating common genetic variants into breast cancer risk prediction models has predominantly been investigated in women of European descent. Previous efforts in Asian studies thus far have focused on the development of Asian-specific PRS, and have been limited by small sample size. Given the difficulties of defining population-specific PRS, a more practical question is whether the PRS developed using data from women of European ancestry is predictive of risk for women of Asian ancestry. In this study, using the largest available data of

Asian women, we independently evaluated the predictive performance of PRS developed based on 287 variants.

Our study showed that the European-ancestry PRS was predictive of overall breast cancer risk for Asians. The magnitudes of association were generally consistent across the ten participating case–control Asian studies and the prospective Singaporean Chinese study. The association was also consistent across the three ethnic groups in Malaysia and Singapore, suggesting that the PRS is associated with similar relative risk estimates in all three ethnicities, though the confidence intervals for Malays and Indians are wide.

The estimated effect size and AUC of both the 287-SNP PRS and 229-SNP PRS were slightly lower than that observed in women of European ancestry. We evaluated the individual association of the 287 SNPs with overall breast cancer risk in Chinese, Malays and Indians separately and compared with the effect sizes in women of European ancestry (Supplementary Data 1). The intraclass correlation coefficients (ICC), taking into account standard errors of estimates, was estimated to be >0.7 for all ethnicities. These results indicate that the susceptibility variants in both populations are largely similar and confer similar relative risks, the lower effect size and AUC may arise from different patterns of linkage disequilibrium. Notably, our analyses showed that the Asian-specific PRS which included only five Asian-specific SNPs[27], achieved AUC of 0.562 (Table 6), suggesting the development of more accurate PRSs in the Asian population is possible when larger cohorts of Asians becomes available to identify population-specifc SNPs.

The mean for the 287-SNP PRS was markedly higher in Asian populations than European populations, but the SD was slightly lower in Asians than Europeans. The lower variation (SD) may reflect the different allele frequency distributions: of the 287 SNPs that are common in women of European ancestry (minor allele frequency > 0.05), 43 are rare in Asian women and therefore contribute minimally to the PRS. In this paper, we have standardised the PRS to the European SD to enable comparison of the performance of the PRS in European and Asian populations. A more relevant approach is to standardised the PRS to the Asian SD, in which case the overall breast cancer OR per unit increase in PRS would be decreased to 1.48 (95% CI: 1.44–1.52). Taken together, these results highlight the need to calibrate the PRS distribution to enable risk models developed based on one population (e.g. Europeans) to be used in another population (e.g. Asians).

## Table 5 Absolute risk of developing overall breast cancer by percentiles.

| Percentile (%) | OR | Chinese Lifetime risk | Age[a] | Malay Lifetime risk | Age[a] | Indian Lifetime risk | Age[a] |
|---|---|---|---|---|---|---|---|
| <1 | 0.38 | 0.02 | NR | 0.02 | NR | 0.02 | NR |
| 1–5 | 0.48 | 0.03 | NR | 0.02 | NR | 0.02 | NR |
| 5–10 | 0.54 | 0.03 | NR | 0.03 | NR | 0.03 | NR |
| 10–20 | 0.67 | 0.04 | NR | 0.03 | NR | 0.03 | NR |
| 20–40 | 0.83 | 0.05 | NR | 0.04 | NR | 0.04 | NR |
| 40–60 | 1 | 0.06 | NR | 0.05 | NR | 0.05 | NR |
| 60–80 | 1.2 | 0.07 | NR | 0.06 | NR | 0.06 | NR |
| 80–90 | 1.51 | 0.09 | 44 | 0.08 | NR | 0.07 | NR |
| 90–95 | 1.82 | 0.11 | 40 | 0.09 | 43 | 0.09 | 46 |
| 95–99 | 2.22 | 0.13 | 37 | 0.11 | 38 | 0.11 | 31 |
| >99 | 2.72 | 0.16 | 35 | 0.13 | 35 | 0.13 | 39 |

Absolute risks were calculated based on self-declared ethnicity and ethnic-specific incidence and mortality data in Singapore and using 287-SNP PRS relative risk for overall breast cancer. *NR* never reached, i.e., the 10-year absolute risk in this percentile never exceed 2.3%.
[a]Age at which 10-year absolute risk exceeds 2.3%. The 2.3% threshold is the average 10-year absolute risk for a 50 years old woman of European ancestry (50 years old is the recommended age to begin regular mammographic screening Singapore).

## Table 6 Association between Asian-specific PRSs and overall breast cancer risk.

| PRS | SNP selection | Number of SNPs published | Number of SNPs used in analyses | SD controls | OR per SD (95% CI) | AUC |
|---|---|---|---|---|---|---|
| Low et al.[27] | Identified in Asian | 5 | 5 | 0.292 | 1.25 (1.23–1.27) | 0.563 |
| Lee et al.[26] | Identified in European/Asian | 51 | 51 | 0.469 | 1.28 (1.26–1.30) | 0.562 |
| Wen et al.[15] | Identified in European/Asian | 44 | 44 | 0.400 | 1.41 (1.39–1.44) | 0.586 |
| Hsieh et al.[17] | Identified in European/Asian | 6 | 6 | 0.356 | 1.10 (1.08–1.11) | 0.533 |
| Chan et al.[19] | Identified in European/Asian | 46 | 45[a] | 0.983 | 1.21 (1.19–1.23) | 0.558 |
| Mavaddat et al.[7] | Identified in European | 313 | 287 | 0.564[b] | 1.51 (1.22–1.86)[b] | 0.617[b] |

PRSs were constructed using per allele log odds ratios as reported in the literature. As Asian case–control studies genotyped by iCOGs array and 744 samples from MYBRCA batch 1 were used as part of the development studies in Wen et al. (2016), to avoid upward bias, we restricted these evaluation analyses to Asian cases–controls studies genotyped using the OncoArray and removed overlapping samples in MYBRCA batch 1.
[a]One SNP rs146699004 was not imputed and hence not included in the analyses.
[b]Analyses of 287-SNP PRS was repeated using the same dataset as described.

The 287-SNP PRS had a lower predictive performance for overall breast cancer among Asians from the three North American studies, compared to the Asian or European studies (Table 2). This somewhat surprising observation might be due to chance, but might reflect a greater admixture with non-Asian ancestry populations, or a greater variation in the distribution of lifestyle factors[26] leading to a greater variation in risk of breast cancer. Larger studies of Asian women in non-Asian countries are needed to provide more reliable estimates.

For subtype analyses using ER-specific PRS, we observed greater discrimination for ER-positive than ER-negative disease. This difference was also seen in European studies, and reflects the fact that the majority of risk SNPs are more strongly associated with ER-positive than ER-negative disease.

The majority of breast cancer studies have been conducted in populations of European descent and, as a result, the screening guidelines for Asian women are often based on those developed in Europe or North America[28,29]. In high income countries with predominantly women of European descent, personalised screening strategy based on age and PRS rather than age alone could reduce the number of people eligible for screening[30], thus potentially reducing overdiagnosis, overtreatment and false-positive diagnoses, which could lead to anxiety and stress in women who have gone for screening[14]. In the Asian context, however, a more cogent argument for stratified screening is to target limited screening resources on those women most likely to benefit. Based on the OR estimated in our analyses, and assuming that a 10-year absolute risk threshold of 2.3% is an appropriate threshold for screening, the majority of Asian women living in the

Asian country with the highest population risk of breast cancer (Singapore) would never reach this threshold (Table 5; Supplementary Fig. 2). Notably, only ~25% of Chinese women, ~16% of Malay women and ~17% of Indian women, would reach this threshold at any point in their lives. It is important to note, however, that Asians will experience a substantial increase in breast cancer incidence over the next decade, and it will therefore be necessary to revisit the screening recommendations over time. To explore this, we simulated the 10-year absolute breast cancer risk of Chinese women using Australian breast cancer incidence[31], which is about twice of that in Singapore (Supplementary Fig. 4). Assuming the breast cancer ORs associated with the PRS remain similar to those estimated here, those who are in the 60–80th percentile of the risk distribution, which would be classified as a low-risk group for screening based on current incidence, would reach the risk threshold for screening at age 45 based on the increased incidences. If the incidence rate reaches that of Western European countries, a similar proportion of women (~20%) would not meet screening threshold at any age[7].

Our study has some limitations worth noting. Although we used the largest dataset of Asian women available to date to evaluate the performance of PRS, the sample size was still too limited to provide precise relative risk estimates for the extremes of the PRS distribution, particularly for ER-specific disease. The majority of the data in the BCAC dataset were generated with the OncoArray, however, ~27% samples were genotyped using iCOGS array, which has lower genome-coverage. Of the 287 SNPs, 42 SNPs have imputation score between 0.75 and 0.9, while 53 SNPs have imputation score below 0.75 in the iCOGs dataset.

This may explain in part the evidence for some heterogeneity in effect sizes between iCOGS and OncoArray datasets. The attenuation (10%) in the effect size of family history of breast cancer on breast cancer risk after adjusting for the 287-SNP PRS is consistent with the predicted contribution of the SNPs to the twofold familial risk of breast cancer for 287 SNPs (~11%, based on an overall OR per Asian SD of 1.48[8]). It is important to note, however, that the estimated association of family history on breast cancer risk (OR = 1.35) is lower compared with other studies (OR = 1.8–3.9 in European studies[32–34] and OR = 1.52–2.1 in Asian studies[16,26]). This might be due to inaccuracies in the family history data. The control women in the largest study (MyBrCa) contributing to these analyses, accounted for ~30% of the total data, were recruited through opportunistic screening which may be enriched for family history relative to the cases. In addition, there was evidence of heterogeneity ($I^2 = 66.1\%$, $p$ value < 0.0001 [chi-squared test]) in the effect sizes of association between family history and breast cancer risk across Asian studies.

In summary, we have shown that a PRS based on common breast cancer susceptibility variants identified in women of European ancestry is a strong predictor of breast cancer risk in Asian women. Furthermore, even though Asians are genetically diverse, our study shows that the PRS derived from women of European ancestry work equivalently well across the diverse ethnic groups in Asia. In the meantime, the PRS developed using data from large European-ancestry studies (providing this is recalibrated to the Asian population being tested) may be used as the basis for Asian-specific breast cancer risk prediction models that include the PRS as well as other predictors of breast cancer risk. These models will allow for higher levels of risk stratification to be achieved, as recently demonstrated in women of European ancestry[10]. Such risk assessment tools could help in resource planning, especially in low- and middle-income countries where resources are limited and population-based screening is unavailable, to improve the efficiency of personalised screening.

## Methods

**Study populations.** The study participants were 45,233 women of Asian ancestry from three sources: (a) 32,238 women (15,755 invasive cases and 16,483 controls) participating in 10 Asian studies in Breast Cancer Association Consortium (BCAC); (b) 2719 women (1507 invasive cases and 1212 controls) of Asian ancestry participating in 3 north American population-based case–control studies in BCAC; and (c) 10,266 women of Chinese ethnicity participating in Singapore Chinese Health Study (SCHS[32,33]). SCHS is a population-based prospective cohort study. Of the total of 10,255 women aged 43–75 years who had not had any cancer diagnosis prior to recruitment, 413 registry-confirmed breast cancers developed over 195,317.2 person years of prospective follow-up. Follow-up started 6 months after recruitment and was censored at age of breast cancer diagnosis, age at last known non-breast cancer status, or age on 31 December 2015, whichever came first. Supplementary Table 1 shows study design and number of breast cancer cases and controls for individual studies. Comparative results for European women were obtained from (a) 4926 cases and 4979 controls from 26 population-based case–control studies participating in BCAC and included in the validation analysis in Mavaddat et al.[7] and (b) ten nested case–control studies within prospective cohorts in BCAC, comprising 11,225 cases and 17,788 controls, included in the test dataset in Mavaddat et al.[7], but excluding subjects <80 years old and for whom age was unknown. All studies were approved by the relevant institutional ethics committees and review boards, and all participants provided written informed consent.

**Genotyping methods.** All samples in BCAC studies were genotyped using one of two arrays: the ~211,155-SNP iCOGS array and the ~533,000-SNP OncoArray[34]. Genotype calling, quality control procedure and imputation has been discarded previously[20,21]. Briefly, samples found to be genotypically not female, discordant or cryptic duplicate pairs, and samples with assay call rate <95% and extreme heterozygosity (<5% or >40%, 4.89 SD from the mean for the ethnicity), were excluded. For first-degree relative pairs, the control was removed from the case–control pairs; otherwise the sample with the lower call rate was excluded. SNPs with assay call rate <95% and deviation from Hardy–Weinberg equilibrium in controls at $p < 10^{-7}$ in controls or $p < 10^{-12}$ for cases were excluded. The iCOGS

and OncoArray datasets were imputed separately using a two-stage imputation approach, using SHAPEIT2[35] for phasing and IMPUTE2[36] for imputation, with 1000 Genomes Project (Phase 3) data as the reference panel[37].

Samples in the prospective study (SCHS) were genotyped using Illumina Global Screening Array. Samples with call rate < 95% and extremes in heterozygosity were excluded. For first- and second-degree relative pairs, the sample with the lower call rate was excluded. Data were imputed using IMPUTE2 with 1000 Genomes Project (Phase 3) as reference panel. Only non-monomorphic SNPs in East Asian population in the reference panel were imputed.

Post-imputation quality was based on the imputation accuracy score INFOSCORE as provided by IMPUTE2[36]. This metric takes values between 0 and 1, with higher values indicating higher imputation certainty and 1 implying perfect imputation.

Principal components analyses were used to identify ethnic outliers and define ancestry informative covariates. For the BCAC data, continental ancestry was derived by combining the data with the 1000 Genomes Project reference data[34]. Individuals with >40% estimated East Asian ancestry are retained. In the second stage, principal components were generated on the Asian ancestry individuals using a subset of uncorrelated SNPs. Similar ancestry informative principal components were generated on the SCHS dataset.

**Statistical methods.** The analyses were based on the 313-SNP PRS developed in women of European ancestry[7]. SNPs with an imputation accuracy score <0.9, based in the MyBrCa and SGBCC studies, combined, were excluded; to ensure accurate determination of PRSs in the ethnic-specific analyses.

We derived PRS for overall breast cancer using Eq. (1)

$$\mathrm{PRS}_{\mathrm{overall}} = \beta_1 x_1 + \beta_2 x_2 + \cdots + \beta_k x_k + \cdots + \beta_n x_n, \quad (1)$$

where $x_k$ is the dosage of risk allele (0–2) for SNP $k$ and $\beta_k$ is the corresponding weight. To avoid bias due to overfitting, we used the weights previously derived for women European ancestry[7]. The ER-specific PRSs (denoted as $\mathrm{PRS}_{\mathrm{ER+}}$ for ER-positive PRS and $\mathrm{PRS}_{\mathrm{ER-}}$ for ER-negative PRS) used same set of SNPs but weights from the hybrid method as reported in Mavaddat et al.[7]; the hybrid method assigns subtype-specific weights to a subset of SNPs for which the effect sizes differ significantly by subtype. The list of SNPs and the corresponding weight are provided in Supplementary Data 1. To enable direct comparison of the performance of each PRS with those reported in European women, we standardised the PRSs by dividing the PRSs of each individual by the SD) of the PRSs in the control subjects from the population-based case–control series in European studies.

Logistic regression models were used to estimate ORs for the association between the standardised PRSs and breast cancer risk. The overall breast cancer PRS was used as predictor in association analyses between overall breast cancer and PRS while for subtype-specific analyses, ER-specific PRS were used as predictors. The PRS were treated as either a continuous or categorical predictor in the model. When used as a categorical variable, the PRS was categorised into the following PRS percentile ranges based on the PRS distribution in controls: 1%, 1–5%, 5–10%, 10–20%, 20–40%, 40–60%, 60–80%, 80–90%, 90–95%, 95–99% and 99–100%. The 40–60% category was used as the reference. For ethnic-specific analyses, analyses were stratified by ethnicity (Chinese, Malay and Indian) using only the MyBrCA and SGBCC datasets. All models were adjusted for first ten principal components and study/array/batch; here samples from the same study that were genotyped in two batches (as was the case for MyBrCa and SGBCC) or on both arrays were treated as different strata for the purposes of adjustment. A Cox proportional hazard model was used for the evaluation of the PRS association with overall breast cancer risk in the prospective cohort and HRs per SD of the PRS were estimated.

The discriminatory accuracy of models for predicting breast cancer risk was evaluated using the area under the receiver operating characteristic curve (AUC), adjusted by study. Estimated ORs by PRS quantiles were compared with the predicted ORs under the model in which the PRS is considered as a continuous covariate and the log (OR) is linearly related to the PRS. To determine the proportion of the familial breast cancer risk that could be explained by PRS, we estimated the OR for the association of first-degree family history and breast cancer risk first adjusted for first 10 principal components and study/array/batch, and then additionally adjusted for the PRS.

To evaluate the effect modification of the PRS (as a continuous covariate) by age and family history of breast cancer in first-degree relatives, we included additional interaction terms in the logistic regression model.

The predicted proportion of the familial relative risk of breast cancer explained by the PRS was estimated by noting that the familial relative risk to first degree relatives of affected individuals due to PRS alone is estimated to be $\lambda_P = \exp(\frac{\gamma^2}{2})$, where $\gamma$ is the OR per one SD (equivalent to the SD of the polygenic risk distribution)[38]. The proportion of the familial relative risk (on a log scale) due to the PRS was therefore estimated by using Eq. (2):

$$\frac{\ln(\lambda_P)}{\ln(\lambda)} = \hat{\gamma}^2 / 2\ln(\lambda), \quad (2)$$

where $\lambda$ is the familial relative risk of breast cancer in first degree relatives, assumed to be 2 for breast cancer.

To compare the effect sizes of individuals SNPs and breast cancer risk with those reported in women of European ancestry, we estimated the effect size of the

association between individual SNP and breast cancer risk in Chinese, Malays and Indians in MyBrCa and SGBCC studies separately using logistic regression, adjusting for age, study and the first 10 principal components, assuming a log-additive genetic model. Intra-class correlation (ICC) was then used to compare the estimated effect sizes with those reported in Mavaddat et al. (2019)[7]. To take into account the sampling error of the effect sizes in the ICC estimate, we fitted a hierarchical model of the form given by Eq. (3):

$$ y_{ij} = \beta_{ij} + \delta_{ij}, \tag{3} $$

where $y_{ij}$ denotes the parameter estimate of SNP $i$ in population $j$, $\beta_{ij}$ are the true parameter estimates and $\delta_{ij} \sim N(0, \sigma_{ij}^2)$ are the sampling errors, with known SDs $\sigma_{ij}$. The model was fitted by using the expectation–maximisation (EM) algorithm[39] in which $\beta_{ij}$ were estimated using a weighted mean of the observed estimates $y_{ij}$ and the group mean $\alpha_i^{(k)}$, as given in Eq. (4)

$$ \hat{\beta}_{ij}^{(k)} = \frac{\frac{\alpha_i^{(k)}}{\sigma_R^2} + \frac{y_{ij}}{\sigma_{ij}^2}}{\frac{1}{\sigma_R^2} + \frac{1}{\sigma_{ij}^2}}, \tag{4} $$

in the E-step and the estimated $\beta_{ij}$ were treated as complete data in the M-step to estimate $\alpha_i^{(k+1)}$ and $\sigma_R^2$, the within-group variance. This process is iterated until the estimated ICC converged.

The age-specific absolute risks of developing breast cancer, adjusting for competing mortality, in each PRS percentile was calculated using Eq. (5)

$$ \mathrm{AR}_g(t) = \sum_{u=0}^{t} \lambda_g(u) \cdot S_g(u) \cdot S_m(u), \tag{5} $$

where $\lambda_g(u)$ is the breast cancer incidence associated with PRS at age $u$, $S_g(u)$ is the probability of being breast cancer free at age $u$, and $S_m(u)$ is the probability of not dying from a cause other than breast cancer to age $u$. The PRS-specific breast cancer incidences, $\lambda_g(u)$, were calculated iteratively by assuming that the average age-specific breast cancer incidence over all PRS percentiles agreed with the population breast cancer incidence[6]. We calculated lifetime and 10-year absolute risks using Singaporean mortality and breast cancer incidence for Chinese, Malays and Indians[23,24]. The recommended screening age at 50 years old in many Asian countries is based on European or North American guidelines[29] and the average 10-year risk of breast cancer for women of European ancestry at age 50 years old is 2.3%[25]. Hence, we determined the proportion of women in the general population who would have the 10-year risk of breast cancer above this threshold, using method as described in Pharoah et al.[38]. To do this, the maximum 10-year absolute risk, adjusting for competing mortality, for women age 20–70, was calculated for each PRS centile category (0–0.1%, …, 99.9–100%), assuming an OR per 1 SD of the PRS of 1.48 (the estimated effect size in Asian studies).

We compared the predictive performance of the European ancestry-based PRS with PRSs that were previously developed or evaluated in Asian populations. The five Asian ancestry-derived PRSs included 5 SNPs[15], 51 SNPs[17], 44 SNPs[19], 6 SNPs[26] and 46 SNPs[27]. The PRSs were derived using Eq. (1) and the corresponding weights reported in the literature. The list of SNPs and corresponding weights are tabulated in Supplementary Table 6.

All statistical analyses were conducted using R v.3.0.3 or Stata v.14.2. Logistic regression and AUC were done using *logistic()* and *comproc()* in Stata, Cox proportional hazard model was done using *Coxph()* in R.

**Reporting summary**. Further information on research design is available in the Nature Research Reporting Summary linked to this article.

## Data availability

Summary statistics (OR and confidence limits) for all SNPs used in the analysis are provided in Supplementary Data 1 of the paper. Request for access to individual level data on which these analyses were based can be made via the Data Access Coordinating Committee of BCAC (BCAC Coordinator: BCAC@medschl.cam.ac.uk). The remaining data are available within the Article, Supplementary Information or available from the authors upon reasonable request. Source data are provided with this paper.

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

## Acknowledgements

This study was supported by grants from Newton-Ungku Omar Fund [grant No. MR/P012930/1] and Wellcome Trust [grant No. v203477/Z/16/Z]. The Malaysian Breast Cancer Genetic Study was established using funds from the Malaysian Ministry of Science, and the Malaysian Ministry of Higher Education High Impact Research Grant [grant No. UM.C/HIR/MOHE/06]. The Malaysian Mammographic Density Study was established using funds raised through the Sime Darby LPGA tournament and the High Impact Research Grant. Additional funding was received from Yayasan Sime Darby, PETRONAS, Estee Lauder Group of Companies and other donors of Cancer Research Malaysia. SGBCC is funded by NUS Start Up Grant, National University Cancer Institute Singapore (NCIS) Centre Grant, NMRC Clinical Scientist Award, NMRC Clinician Scientist Award-Senior Investigator, Asian Breast Cancer Research Fund and Breast Cancer Prevention Programme under Saw Swee Hock School of Public Health. Recruitment of controls were funded by the Biomedical Research Council (05/1/21/19/425). W.K.H. is the recipient of L'Oreal-UNESCO For Women in Science National Fellowship. J.L. is the recipient of a National Research Foundation Singapore Fellowship (NRF-NRFF2017-02). A.C.A. is supported through Cancer Research—UK (C12292/A20861). J.S. holds a Canada Research Chair in Oncogenetics. The PERSPECTIVE I&I project is funded by the Government of Canada through Genome Canada and the Canadian Institutes of Health Research, the Ministère de l'Économie et de l'Innovation du Québec through Genome Québec, the Quebec Breast Cancer Foundation, the CHU de Quebec Foundation and the Ontario Research Fund. Genotyping of the OncoArray was principally funded from three sources: the PERSPECTIVE project, funded by the Government of Canada through Genome Canada and the Canadian Institutes of Health Research, the 'Ministère de l'Économie, de la Science et de l'Innovation du Québec' through Genome Québec, and the Quebec Breast Cancer Foundation; the NCI Genetic Associations and Mechanisms in Oncology (GAME-ON) initiative and Discovery, Biology and Risk of Inherited Variants in Breast Cancer (DRIVE) project (NIH Grants U19 CA148065 and X01HG007492); and Cancer Research UK (C1287/A10118 and C1287/A16563). BCAC is funded by Cancer Research UK (C1287/A16563), by the Caucasian Community's Seventh Framework Programme under grant agreement 223175 (HEALTH-F2-2009-223175) (COGS) and by the Caucasian Union's Horizon 2020 Research and Innovation Programme under grant agreements 633784 (B-CAST) and 634935 (BRIDGES). Combining of the GWAS data were supported in part by The National Institute of Health (NIH) Cancer Post-Cancer GWAS initiative grant U19 CA 148065 (DRIVE, part of the GAME-ON initiative). For MyBrCa, we want to thank Pui-Yoke Kwan, Sean Wen, Norhashimah Hassan, Peter Choon-Eng Kang, In-Nee Kang, Kah-Nyin Lai, Hanis Hasmad, Jin-Tong Ng, Gaik-Theng Toh, Nancy Geen-See Tan, Suhaida Selamat, Nancy Geen-See Tan, Rohaya Mohd Kasim, Malkit Kaur Dhillon, Thin-Chai Liu, Ernie Azwa, Hanani Che Halim, Leelavathy Krishnan, Don-Na Tan, Sweet-Lin Goh, Nur Naquiah Kamaruddin, Faridah Bakri, the participants of this study, and all staff at Cancer Research Malaysia, University Malaya, and Sime Darby Medical Centre who assisted in recruitment and interviews. For SGBCC, we want to thank the programme manager Jenny Liu, clinical research coordinators/research assistants Siok-Hoon Yeo, Kimberley Chua, Ting-Ting Koh, Amanda Ong, Jin-Yee Lee, Michelle Mok, Ying-Jia Chew, Jing-Jing Hong and Hui-Min Lau for their contributions in recruitment and Alexis Khang for processing the DNA samples. We also want to thank all the participants' support to SGBCC. There are additional funding and acknowledgements listed in Supplementary Note 1.

## Author contributions

Study design: W.K.H., N.M., J. Simard, D.F.E., S.H.T. and A.C.A.; writing group: W.K.H., M.M.T., N.M., M.C.T., J.L, D.F.E. and S.H.T. and A.C.A.; data management: M.M.T., M.K.B. and Q.W.; bioinformatic analysis: J.D., J.P.T. and K. Michailidou; Statistical analysis: W.K.H., N.M, M.M.T., M.C.T., S.M., P.J.H. and D.F.E.; provided DNA samples and/or phenotypic data: S.M., J.L., D.K., J.Y.C., S.J., X.O.S., S.Y.Y., S.K.P., S.W.K., C.Y.S., J.C.Y., E.Y.T., P.M.Y.C., K. Muir, A.L., A.H.W., D.O.S., K. Matsuo, H.I., C.W.C., J.N., W.S.Y., S.H.L., G.H.L., A.K., T.L.C., S.M.T., J. Seah, E.M.J., A.W.K., W.P.K., C.C.K., M.I., T.Y., K.M.V.T., K.T.B.T., J.J.S., K.J.A, S.N.H., K.R., A.V., X.S., P.D.P.P., W.Z., A.M.D., J. Simard, R.M.v.D., C.H.Y., N.A.M.T., M.H. and S.H.T. All authors read and approved the final version of the paper.

## Competing interests

The authors declare no competing interests.

## Additional information

Weang-Kee Ho [1,2✉], Min-Min Tan [1,2], Nasim Mavaddat[3], Mei-Chee Tai[2], Shivaani Mariapun[1,2], Jingmei Li [4,5], Peh-Joo Ho [4], Joe Dennis [3], Jonathan P. Tyrer [6], Manjeet K. Bolla[3], Kyriaki Michailidou[3,7,8], Qin Wang[3], Daehee Kang[9,10,11], Ji-Yeob Choi[10,11], Suniza Jamaris[12], Xiao-Ou Shu[13], Sook-Yee Yoon[2], Sue K. Park [9,10,11], Sung-Won Kim[14], Chen-Yang Shen[15,16], Jyh-Cherng Yu[17], Ern Yu Tan[18], Patrick Mun Yew Chan[18], Kenneth Muir [19], Artitaya Lophatananon[19], Anna H. Wu[20], Daniel O. Stram [20], Keitaro Matsuo [21,22], Hidemi Ito [21,22], Ching Wan Chan [23,24], Joanne Ngeow[25,26], Wei Sean Yong[27],

Swee Ho Lim[28], Geok Hoon Lim [28], Ava Kwong[29,30,31], Tsun L. Chan[29,32], Su Ming Tan[33], Jaime Seah[33], Esther M. John[34], Allison W. Kurian [34,35], Woon-Puay Koh[36,37], Chiea Chuen Khor [4], Motoki Iwasaki[38], Taiki Yamaji[38], Kiak Mien Veronique Tan[27,39], Kiat Tee Benita Tan[27,39], John J. Spinelli[40,41], Kristan J. Aronson[42], Siti Norhidayu Hasan[2], Kartini Rahmat [43], Anushya Vijayananthan[43], Xueling Sim [37], Paul D. P. Pharoah [3,44], Wei Zheng[13], Alison M. Dunning [44], Jacques Simard [45], Rob Martinus van Dam[37,46], Cheng-Har Yip [47], Nur Aishah Mohd Taib[12], Mikael Hartman[5], Douglas F. Easton [3,44,48], Soo-Hwang Teo [2,12,48✉] & Antonis C. Antoniou[3,48]

[1]School of Mathematical Sciences, Faculty of Science and Engineering, University of Nottingham Malaysia, Jalan Broga, Semenyih 43500 Selangor, Malaysia. [2]Cancer Research Malaysia, 1 Jalan SS12/1A, Subang Jaya 47500 Selangor, Malaysia. [3]Centre for Cancer Genetic Epidemiology, Department of Public Health and Primary Care, University of Cambridge, CB1 8RN, Cambridge, UK. [4]Human Genetics, Genome Institute of Singapore, 60 Biopolis St, 138672 Singapore, Singapore. [5]Department of Surgery, National University Hospital and NUHS, 1E Kent Ridge Road, 119228 Singapore, Singapore. [6]Strangeways Research Laboratory, University of Cambridge, 2 Worts' Causeway, Cambridge, UK. [7]Biostatistics Unit, The Cyprus Institute of Neurology & Genetics, 6 Iroon Avenue, 2371 Ayios, Dometios, Cyprus. [8]Cyprus School of Molecular Medicine, The Cyprus Institute of Neurology & Genetics, 6 Iroon Avenue, 2371 Ayios, Dometios, Cyprus. [9]Department of Preventive Medicine, Seoul National University College of Medicine, 103 Daehak-Ro, Jongno-Gu, 03080 Seoul, Korea. [10]Department of Biomedical Sciences, Seoul National University Graduate School, 103 Daehak-Ro, Jongno-Gu, 03080, Seoul, Korea. [11]Cancer Research Institute, Seoul National University, 103 Daehak-Ro, Jongno-Gu, Seoul 03080, Korea. [12]Department of Surgery, Faculty of Medicine, University of Malaya, Jalan Universiti, 50630 Kuala Lumpur, Malaysia. [13]Division of Epidemiology, Department of Medicine, Vanderbilt Epidemiology Center, Vanderbilt-Ingram Cancer Center, Vanderbilt University School of Medicine, 1161 21st Ave S # D3300, Nashville, TN 37232, USA. [14]Department of Surgery, Daerim Saint Mary's Hospital, 657 Siheung-Daero, Daerim-Dong, Yeongdeungpo-Gu, 07442 Seoul, Korea. [15]Institute of Biomedical Sciences, Academia Sinica, 115128, Section 2, Academia Road, Taipei, Taiwan. [16]School of Public Health, China Medical University, Taichung, Taiwan. [17]Department of Surgery, Tri-Service General Hospital, Taipei 114, Taiwan. [18]Department of General Surgery, Tan Tock Seng Hospital, Singapore 308433, Singapore. [19]Division of Population Health, Health Services Research and Primary Care, School of Health Sciences, The University of Manchester, Oxford Road, M13 9PL, Manchester, UK. [20]Department of Preventive Medicine, Keck School of Medicine, University of Southern California, 1975 Zonal Ave, Los Angeles 90033 CA, USA. [21]Division of Cancer Epidemiology and Prevention, Aichi Cancer Center Research Institute, 1-1 Kanokoden, Chikusa-Ku, 464-8681 Nagoya, Japan. [22]Division of Cancer Epidemiology, Nagoya University Graduate School of Medicine, 65 Tsurumai-Cho, Showa-Ku, 466-8550 Nagoya, Japan. [23]Department of Surgery, Yong Loo Lin School of Medicine, National University of Singapore, 117597 Singapore, Singapore. [24]National University Hospital, National University Health System, Singapore, Singapore. [25]Division of Medical Oncology, National Cancer Centre Singapore, 11 Hospital Drive, 169610 Singapore, Singapore. [26]Oncology Academic Clinical Program, Duke-NUS Graduate Medical School, 8 College Road, 169857 Singapore, Singapore. [27]Division of Surgical Oncology, National Cancer Centre, Singapore, Singapore. [28]Breast Department, KK Women's and Children's Hospital, Singapore100 Bukit Timah Road, 229899, Singapore. [29]Hong Kong Hereditary Breast Cancer Family Registry, Cancer Genetics Centre, 18A Kung Ngam Village Road, Happy Valley, Hong Kong. [30]Department of Surgery, The University of Hong Kong, 102 Pokfulam Road, Pok Fu Lam, Hong Kong. [31]Department of Surgery, Hong Kong Sanatorium and Hospital, 2 Village Rd, Happy Valley, Hong Kong. [32]Department of Pathology, Hong Kong Sanatorium and Hospital, 2 Village Rd, Happy Valley, Hong Kong. [33]General Surgery, Changi General Hospital, Singapore, Singapore. [34]Department of Medicine, Division of Oncology, Stanford Cancer Institute, Stanford University School of Medicine, 780 Welch Road, Suite CJ250C, Stanford 94304 CA, USA. [35]Department of Health Research and Policy—Epidemiology, Stanford University School of Medicine, 259 Campus Drive, Stanford 94305 CA, USA. [36]Health Services and Systems Research, Duke-NUS Medical School, Stanford University School of Medicine, 8 College Road, 169857 Singapore, Singapore. [37]Saw Swee Hock School of Public Health, National University of Singapore and National University Health System, 12 Science Drive 2, #10-01, 117549 Singapore, Singapore. [38]Division of Epidemiology, Center for Public Health Sciences, National Cancer Center, 5-1-1 Tsukiji, Chuo-Ku, 104-0045 Tokyo, Japan. [39]Department of General Surgery, Singapore General Hospital, Singapore, Singapore. [40]Population Oncology, BC Cancer, 675 West 10th Avenue, VancouverV5Z 1G1 BCCanada. [41]School of Population and Public Health, University of British Columbia, 2329 West Mall VancouverV6T 1Z4 BCCanada. [42]Department of Public Health Sciences, and Cancer Research Institute, Queen's University, 10 Stuart Street, Kingston K7L 3N6 ON, Canada. [43]Biomedical Imaging Department, Faculty of Medicine, University of Malaya, Kuala Lumpur, Malaysia. [44]Centre for Cancer Genetic Epidemiology, Department of Oncology, University of Cambridge, 2 Worts' Causeway, CB1 8RN, Cambridge, UK. [45]Genomics Center, CHU de Québec-Université Laval Research 2705 Blvd Laurier Québec (Québec) G1V 4G2, Quebec, Canada. [46]Departments of Medicine, Yong Loo Lin School of Medicine, National University of Singapore, Singapore, Singapore. [47]Sime Darby Medical Centre, 1 Jalan SS12/1A, Subang Jaya 47500 Selangor, Malaysia. [48]These authors contributed equally: Douglas F. Easton, Soo-Hwang Teo, Antonis C. Antoniou. ✉email: weangkee.ho@nottingham.edu.my; soohwang.teo@cancerresearch.my

