## [Peer Review File · Nature Communications]

Reviewers' comments:

Reviewer #1 (Remarks to the Author):

This manuscript presents a very large and fairly comprehensive analysis studying the application of a polygenic risk score derived using data from European women to characterize risk in Asian women. The components of the paper are well developed but the actual presentation has many issues and the paper needs a great deal of revision for clarity. I have many minor comments and two more major concerns.

Major question – it is not very clear that genetic analysis to identify 1% of women at polygenic risk would have clinical utility or be feasible. If the PRS is set to the top 5% does that provide an Odds ratio that is meaningful or a prospective risk that would lead to different management of women? Some sort of sensitivity analysis considering at least an upper 5% threshold and perhaps an upper 10% threshold should be included.

What is the AUC when you include versus exclude Family History. Table 3 gives the attenuation in OR when including family history but the more interesting element to me would be to evaluate what the AUC is when you include markers and Family history.

I think it would be useful to also plot the distribution of risk according to quantiles of risk and according to age. This would better illustrate the lack of information from individuals at low PRS across all ages.

Minor comments: Line 143. Should read 'mutations in these genes are rare' not these genes are rare, since almost everyone has two copies of these genes (except rare people with deletions). Line 157 – this discussion is not very comprehensive. While it may be true that PRS has some utility how much does it improve beyond traditional and early collect information like family history?

Lines 192-197 it would be helpful to have some information about what the interpretation of the PRS mean score would be. Is this score somehow standardized? If not then should it be to make the interpretation more straightforward?

Lines 226-230 gives changes in percent attenuation without indicating what is being attenuated or what the relevance of this sentence is. Please clarify this section without requiring close perusal of the table since the sentence makes no sense as written.

Lines 249-251. The actual AUCs should be cited in the text where it is being highlighted.

These sentences do not make sense. Smaller than what? Since this sentence starts a new paragraph there is no referent for that, so the sentence is uninterpretable. 'The OR per SD was significantly smaller ($p < 0.05$) than that in the Asian studies from Asia, 267 for overall breast cancer (1.36, 95% CI = 1.25 – 1.49) and ER-positive breast cancer (1.38, 268 95% CI = 1.25 – 1.53), but significant higher ($p < 0.05$) for ER-negative breast cancer (1.49, 269 95% CI = 1.26 – 1.76, Table 2(b)).'

Around lines 284 – where do the referent risks come from for populations that are being described? It does not sound as if population risks for breast cancer are being derived from cohort studies but rather from a population database but there is no reference so it is impossible to tell how these analyses were conducted.

Line 339 – effect size of PRS - the authors do not define their threshold. Do they mean top 1% of Asian PRS as a cut off? Do they mean top 1% of European?

Line 347 – lower predictive accuracy in Asians- the author need to point to a table in this

discussion and give some values since this information was not previously cited. Or it could be moved to the results and described more fully (which would be preferable).

The discussion cites that a screening regimen based on PRS and age could reduce numbers of women undergoing screening and cites a prior paper but shouldn't they also derive these numbers from their own analysis?

Line 372 to 375 – this presents some interesting observations that have not been described previously and do not cite a figure or table so are uninterpretable.

Lines 381-387- great that simulations were done. Please present the results in a figure and or table. Otherwise this is unacceptable presentation of unsupported information.

Lines 406-410. Did the authors check for heterogeneity in the ORs for family history by study? Opportunistic control selection could definitely lead to enrichment of FH positive controls. That might also yield a similar frequency of FH in case populations and variability in FH+ controls, which could be tested for.

432-435 – the sentence structure in this section is weird and confusing e.g. 'All three studies are population-based case-control studies; and (c) 10,266 women 433 of Chinese ethnicity participating in Singapore Chinese Health Study (SCHS)' In this section you have a period break before All so it is not very clear that you are referring to the prior sentence and then you join this unclear phrase the next unclear phrase after a semicolon. Please break unrelated phrases into sentences and join related clauses to the antecedent sentences.

Line 472 'Post-imputation quality of all studies was based on the IMPUTE imputation quality score.' This should not be a guessing game. Please describe what these imputation quality metrics are or explain them further in a supplementary methods that are referenced.

Lines 485-490 the English is missing plurals and articles so should be revised. It is great to see a reference for how the PRS was standardised, but I think this information should also be placed in the text where the means of PRS are presented.

Line 491-495 – the manuscript indicates that the PRS was stratified into quartiles and the middle quintile was used for referent but that is not very clearly relevant to the paper where top 1% of distribution was used. I think in this section the description is fairly clear but where the 1% cut off is described you need to cite that you are using the middle quintile as referent.

Line 507-509 – AUC analyses were adjusted by study where relevant. This needs to be explained further.

Lines 515-519 please give a reference for why this formula is appropriate.

The references on line 551 should be cited in the main body of the paper as requested above.

For reproducibility it would be better to specify which modules or R or Stata were used for major analyses rather than the very generic reference that is given.

Table 2 needs to cite in the footnote that the cutpoint is top 1% of the PRS in controls if that is what was used for stratifying high risk individuals.

Reviewer #2 (Remarks to the Author):

Ho and colleagues evaluated the performance of a European-derived polygenic risk score (PRS) in an Asian population of 17,262 breast cancer cases and 17,695 controls. Most of these women were living in Asian countries, with a smaller population of Asian-Americans (1,507 cases and 1,212 controls). In addition, they evaluated the PRS performance in 10,255 Chinese women from a prospective cohort (413 incident breast cancers). This is a well-conducted study with findings that have public health relevance since it provides estimates for polygenic breast cancer risk in Asian women that could be used for risk-stratified screening/prevention. The main finding is that the European-derived PRS can provide substantial risk stratification of Asian women, however the stratification is lower than for European-ancestry women. Although this is the largest study conducted in Asians to date, as the authors pointed out, even larger studies are needed to obtain precise OR estimates at the extreme of the PRS distribution, or to further improve the performance of PRS in Asian women.

Below are some comments/questions for consideration:

Q1: The authors used PRS weights based on estimates from European ancestry populations, rather than being re-estimated in Asian populations. This is a reasonable approach as the sample sizes to estimate weights are much larger in Europeans and the authors showed evidence of consistent effect size for genetic variants between European and Asian populations. However, given that previous publications have attempted to derive Asian-specific PRS, it would be of interest to directly compare the prediction performance between previous Asian-specific PRS (derived with lower sample size) and the European based PRS.

Q2: Did the authors consider incorporating Asian-specific risk variants that are not in the 313-PRSs developed for European populations?

Q3: The authors evaluated the PRS based on 313 variants reported in Mavaddat et al. 2019. However, 26 of the 313 variants were excluded due to low imputation accuracy score in Asians using the 1000 genome reference panel, resulting in a 287-variant PRS. This is a limitation since it does not allow direct comparison with the previously published PRS. Did the author consider imputing the genotypes based on larger reference panels such as HRC and TopMed?

Q4: For the Singapore Chinese Health Study (SCHS), only 229 of the 287 variants that were found to be polymorphic and imputable. Although this is a valuable study because of its prospective nature and information on ancestry, it is a bit confusing to show both the 287-SNP PRS and 229-SNP PRS as main findings. I suggest moving the 229-SNP PRS findings to supplement or make clearer what is the added value of these analyses as main findings. I do not see much value added on tables 1 or 2. Findings on Tables 4 and 5 by ancestry are of interest, are PRS means (SD) on Table 4 based on the 229 or 287 variant PRS?

Q5: Are the principal components calculated separately for the 10 Asian studies in BCAC and the three Asian-American studies?

Q6: The authors find a smaller RR for family history than previously reported in European-ancestry studies. Are there previous studies in Asian populations that support this difference?

Q7: The 287-variant PRS was less predictive of breast cancer risk in the three Asian-American studies than in the studies conducted in Asian (which found no differences between Asians of Chinese, Malay or Indian origin). This is a potentially important finding that requires confirmation in larger studies. Could the authors formally test if these differences are statistically significant in the current population? Based on these findings, can the authors comment on what needs to be done to use PRS in Asian populations in the US? Or other countries?

Minor comments:

Q1: In the imputation for BCAC, the phase of the 1000 genome project should be specified.

Q2: The EM algorithm needs citation

Q3: The version of R and Stata needs to be listed

Q4: Indicate population/sample size included in each table/figure or a reference to a table/figure showing the population/sample size.

Q5: Table 1- change format to match Table 2 (with less blank cells).

Q6: Consider adding the European estimates to Figure 2

Q7: Consider adding estimates from American Asians to Figure 3.

Q8: Tables 3 and 4- indicate what PRS is shown – 287-SNP PRS?

RESPONSE TO THE REVIEW COMMENTS

We appreciate the time and effort taken by the reviewers in reviewing the manuscript.

Reviewer #1

This manuscript presents a very large and fairly comprehensive analysis studying the application of a polygenic risk score derived using data from European women to characterize risk in Asian women. The components of the paper are well developed but the actual presentation has many issues and the paper needs a great deal of revision for clarity. I have many minor comments and two more major concerns.

(1) Major question – it is not very clear that genetic analysis to identify 1% of women at polygenic risk would have clinical utility or be feasible. If the PRS is set to the top 5% does that provide an Odds ratio that is meaningful or a prospective risk that would lead to different management of women? Some sort of sensitivity analysis considering at least an upper 5% threshold and perhaps an upper 10% threshold should be included.

Response: Although we particularly highlighted the results for the top 1% of the polygenic risk, our analyses provide estimates for different categories of the PRS. We agree that risks at lower thresholds may also be important.

In our analyses, PRS was treated either as a continuous variable or as a categorical variable. When treated as continuous variable (as shown in Table 2), the odds ratio represents the increased breast cancer risk for every unit increase in PRS. This can be converted to a predicted OR for different percentiles of the PRS distribution; as demonstrated in Figure 2, these predicted ORs agree very well with the observed ORs. The analyses where the PRS was treated as a categorical variable (for example Figure 2 and Supplementary Table 5), the association with breast cancer risk by category, compared to women with median PRS (middle quintile) as the reference group, directly addresses the reviewer's comment. This is described in Result section, last paragraph of page 7.

We also provided absolute risk estimates for different categories of the PRS, for Singaporean women. We showed that Singaporean women in the 99th percentile, 95-99th percentile, 90-95th percentile and 80-90th percentile of the PRS distribution have lifetime absolute risks of breast cancer that range from 9-16% depending on ethnicity. We have added a further sentence on page 10 to emphasise this:

“For women between the 90 and 99th percentiles of the risk distribution, the lifetime risks vary from 9-13%.”

We note that Singaporean Chinese women in the top 25% (top 16% for Malay women, 17% for Indian women) of the PRS distribution will reach the risk threshold of screening at some point in their lives. These results indicate that the appropriate screening recommendations may vary by PRS, and that the appropriate thresholds may be different in different populations. We have

also calculated the proportion of cases accounted for in these fractions of the populations who will reach the screening threshold (results are included in the first paragraph of page 11 and method was included in page 20).

(2) What is the AUC when you include versus exclude Family History. Table 3 gives the attenuation in OR when including family history but the more interesting element to me would be to evaluate what the AUC is when you include markers and Family history.

Response: The AUC for model with PRS alone in the Asian studies was 0.613. There is only a slight improvement in AUC after including family history into the model with PRS (AUC = 0.616). Thus, family history adds very little discrimination, in comparison to the PRS. We added the following sentence on page 8:

“Including family history in the model, in addition to the PRS, increased the AUC only slightly (0.616 vs 0.613 for PRS alone; Table 2), “

(3) I think it would be useful to also plot the distribution of risk according to quantiles of risk and according to age. This would better illustrate the lack of information from individuals at low PRS across all ages.

Response: Due to limited number of cases and controls in the extreme of PRS, it would not be possible to generated results stratified by both quantiles of PRS and age categories, as the reviewer suggests. Instead, we have assessed the effect modification of the PRS by age by including a PRSxage interaction term in the model. The results are given in Supplementary Table 3, and show no evidence of interaction between PRS and age. Thus, our results are based on the most parsimonious model, in which the PRS effect is the same across all ages. To illustrate this, we have also estimated the effect sizes for the PRS, modelled as a continuous variable, in women of different age categories. The estimates are shown in the Supplementary Table 4 and the table below, and demonstrate that the data are consistent with the same OR per SD of the PRS in each category.

Age Group (years)	OR per SD	L95%CI	U95%CI
<40	1.47	1.35	1.59
40-50	1.54	1.47	1.61
50-60	1.53	1.46	1.60
60-70	1.55	1.45	1.65
70-80	1.50	1.31	1.73

Minor comments:

(4) Line 143. Should read 'mutations in these genes are rare' not these genes are rare, since almost everyone has two copies of these genes (except rare people with deletions).

Response: We have edited the entire sentence for clarity and made the suggested change on page 5.

(5) Line 157 – this discussion is not very comprehensive. While it may be true that PRS has some utility how much does it improve beyond traditional and early collect information like family history?

Response: We have expanded the second paragraph of the Introduction to include a brief discussion on the added value of PRS in breast cancer risk prediction model based on classic risk factors, including family history. As noted above, and in the results, family history provides very little risk discrimination, in comparison to the PRS.

(6) Lines 192-197 it would be helpful to have some information about what the interpretation of the PRS mean score would be. Is this score somehow standardized? If not then should it be to make the interpretation more straightforward?

Response: The mean and SD of PRSs in Table 1 were not standardised. This was deliberate - we presented the mean and SD of the raw PRSs to demonstrate that while the SD of PRS distribution in Asian women is only slightly lower than that in Europeans, the mean of the PRS is markedly higher in Asian than European women. This indicates that it is important to calibrate risk models that include the PRS to the population-specific PRS distribution. This was discussed in the Discussion. We have rephrased the sentences in the fourth paragraph of Discussion for clarity. It is worth noting, however, that, in the context of a given population, the mean PRS is not itself relevant, since the results are invariant to adding to a constant.

(7) Lines 226-230 gives changes in percent attenuation without indicating what is being attenuated or what the relevance of this sentence is. Please clarify this section without requiring close perusal of the table since the sentence makes no sense as written.

Response: We have edited the paragraph on page 8 to clarify that we are referring to the attenuation in ORs of family history after adjusting for PRS.

(8) Lines 249-251. The actual AUCs should be cited in the text where it is being highlighted.

Response: Since the analyses involved nine AUCs values, we include the range of AUCs across the three ethnicities for overall and subtype-specific breast cancer in the text instead of listing the individual AUCs values (page 9).

(9) These sentences do not make sense. Smaller than what? Since this sentence starts a new paragraph there is no referent for that, so the sentence is uninterpretable. 'The OR per SD was significantly smaller ($p < 0.05$) than that in the Asian studies from Asia, 267 for overall breast cancer (1.36, 95% CI = 1.25 – 1.49) and ER-positive breast cancer (1.38, 268 95% CI = 1.25 – 1.53), but significant higher ($p < 0.05$) for ER-negative breast cancer (1.49, 269 95% CI = 1.26 – 1.76, Table 2(b)).'

Response: We have rephrased the sentence to clarify that the comparison is between the effect estimates from American and Asian studies (page 9).

(10) Around lines 284 – where do the referent risks come from for populations that are being described? It does not sound as if population risks for breast cancer are being derived from cohort studies but rather from a population database but there is no reference so it is impossible to tell how these analyses were conducted.

Response: We have included references to the sources for Singapore breast cancer incidence and mortality (page 10).

(11) Line 339 – effect size of PRS - the authors do not define their threshold. Do they mean top 1% of Asian PRS as a cut off? Do they mean top 1% of European?

Response: In this analysis, PRS was treated as continuous variable as in the analyses in Table 2 but standardized to the SD of PRS in Asian population. Hence, the reported odds ratio represents the increased breast cancer risk per standard deviation increase in PRS. We have rephrased the relevant paragraph on page 12 to improve clarity.

(12) Line 347 – lower predictive accuracy in Asians- the author need to point to a table in this discussion and give some values since this information was not previously cited. Or it could be moved to the results and described more fully (which would be preferable).

Response: We have cited the Table in the text. The comparison of the effect sizes from Asian Americans and Asians from Asia is also discussed in the Results section under the subtitle “Association between PRS and breast cancer risk in Asian Americans” and has been highlighted for the reviewer’s attention (page 9 and 13).

(13) The discussion cites that a screening regimen based on PRS and age could reduce numbers of women undergoing screening and cites a prior paper but shouldn't they also derive these numbers from their own analysis?

Response: On the last paragraph of page 13, we discuss that, based on the odds ratios estimated in our analyses, and breast cancer incidence and mortality for Singapore, majority

women in Singapore would never reach the 2.3% 10-year risk threshold for screening. This was originally stated in a separate paragraph, for clarity, we have rephrased the relevant sentences and combined the two adjoint paragraphs.

(14) Line 372 to 375 – this presents some interesting observations that have not been described previously and do not cite a figure or table so are uninterpretable.

Response: We have cited Table 5 and Supplementary Figure 2 in this paragraph (page 13).

(15) Lines 381-387- great that simulations were done. Please present the results in a figure and or table. Otherwise this is unacceptable presentation of unsupported information.

Response: We have included the simulation results as Supplementary Figure 4 (page 14).

(16) Lines 406-410. Did the authors check for heterogeneity in the ORs for family history by study? Opportunistic control selection could definitely lead to enrichment of FH positive controls. That might also be yield a similar frequency of FH in case populations and variability in FH+ controls, which could be tested for.

Response: A test of heterogeneity in the ORs for association between family history and breast cancer risk among Asian studies gives $I^2 = 66.1\%$ ($p < 0.0001$). We have included the result of this test in the Discussion and stated that this is a limitation of our study. (page 14)

(17) 432-435 – the sentence structure in this section is weird and confusing e.g. ‘All three studies are population-based case-control studies; and (c) 10,266 women 433 of Chinese ethnicity participating in Singapore Chinese Health Study (SCHS)’ In this section you have a period break before All so it is not very clear that you are referring to the prior sentence and then you join this unclear phrase the next unclear phrase after a semicolon. Please break unrelated phrases into sentences and join related clauses to the antecedent sentences.

Response: We have reworded these sentences as requested (page 15).

(18) Line 472 ‘Post-imputation quality of all studies was based on the IMPUTE imputation quality score.’ This should not be a guessing game. Please describe what these imputation quality metrics are or explain them further in a supplementary method that are referenced.

Response: The imputation quality score is an information metric reported by the imputation software IMPUTE2. Values range from 0 and 1, with a higher number indicating higher imputation accuracy. We have re-written the section on imputation on pages 16-17 to describe the imputation process in more detail. We have also rephrased and moved this sentence to appear earlier in text to further clarify that this is an output from the IMPUTE2 software.

(19) Lines 485-490 the English is missing plurals and articles so should be revised. It is great to see a reference for how the PRS was standardised, but I think this information should also be placed in the text where the means of PRS are presented.

Response: We thank the reviewer for pointing this out. We have corrected the text accordingly. We have also included a description on how PRS was standardized (page 17). When presenting the mean and SD of PRSs (Table 1), unstandardised PRSs were used to perform the comparisons of the different PRSs as described in the response to *comment #6*, so this text is not relevant there.

(20) Line 491-495 – the manuscript indicates that the PRS was stratified into quartiles and the middle quintile was used for referent but that is not very clearly relevant to the paper where top 1% of distribution was used. I think in this section the description is fairly clear but where the 1% cut off is described you need to cite that you are using the middle quintile as referent.

Response: For all the tables and figures that reported analyses where PRSs were treated as categorical variable, we have included in the footnote that middle quintile was used as reference group. We have also re-written the section in the methods to further clarify the percentile categories used (page 18).

(21) Line 507-509 – AUC analyses were adjusted by study where relevant. This needs to be explained further.

Response: We have rephrased the sentence to indicate that AUCs were adjusted for study (removing “where relevant” which is redundant, page 18).

(22) Lines 515-519 please give a reference for why this formula is appropriate.

Response: We have included the reference for the formula and edited the text in the methodology section to include more detail description of the formula (page 19).

(23) The references on line 551 should be cited in the main body of the paper as requested above.

Response: We have included the reference as suggested. See comment #10.

(24) For reproducibility it would be better to specify which modules or R or Stata were used for major analyses rather than the very generic reference that is given.

Response: We have included the libraries and functions that were used in the Method section (page 20).

(25) Table 2 needs to cite in the footnote that the cutpoint is top 1% of the PRS in controls if that is what was used for stratifying high risk individuals.

Response: See response to comment #1.

Reviewer #2

Ho and colleagues evaluated the performance of a European-derived polygenic risk score (PRS) in an Asian population of 17,262 breast cancer cases and 17,695 controls. Most of these women were living in Asian countries, with a smaller population of Asian-Americans (1,507 cases and 1,212 controls). In addition, they evaluated the PRS performance in 10,255 Chinese women from a prospective cohort (413 incident breast cancers). This is a well-conducted study with findings that have public health relevance since it provides estimates for polygenic breast cancer risk in Asian women that could be used for risk-stratified screening/prevention. The main finding is that the European-derived PRS can provide substantial risk stratification of Asian women, however the stratification is lower than for European-ancestry women. Although this is the largest study conducted in Asians to date, as the authors pointed out, even larger studies are needed to obtain precise OR estimates at the extreme of the PRS distribution, or to further improve the performance of PRS in Asian women.

Below are some comments/questions for consideration:

Q1: The authors used PRS weights based on estimates from European ancestry populations, rather than being re-estimated in Asian populations. This is a reasonable approach as the sample sizes to estimate weights are much larger in Europeans and the authors showed evidence of consistent effect size for genetic variants between European and Asian populations. However, given that previous publications have attempted to derive Asian-specific PRS, it would be of interest to directly compare the prediction performance between previous Asian-specific PRS (derived with lower sample size) and the European based PRS.

Response: We have now done the analyses as suggested by the reviewer. We evaluated the performance of five published Asian PRSs in predicting breast cancer risk in our studies, and now show that the European-ancestry based PRS outperformed (by a wide margin) all five previously published Asian PRS (Table 6). We have included these findings in the Results (page 11) and in the Discussion (third paragraph of page 12).

Q2: Did the authors consider incorporating Asian-specific risk variants that are not in the 313-PRSs developed for European populations?

Response: The focus of the current study is on the evaluation of the utility of the European ancestry-based PRS, which have already been extensively validated in European populations, in predicting breast cancer risk in Asian populations. Therefore, we did not incorporate Asian-specific risk variants. This will be part of a future study focused on the development of Asian-specific PRS.

Q3. The authors evaluated the PRS based on 313 variants reported in Mavaddat et al. 2019. However, 26 of the 313 variants were excluded due to low imputation accuracy score in Asians using the 1000 genome reference panel, resulting in a 287-variant PRS. This is a limitation since it does not allow direct comparison with the previously published PRS. Did the author consider imputing the genotypes based on larger reference panels such as HRC and TopMed ?

Response: The data in this study are part of a large consortium effort, in which variants were imputed to the 1000 genomes project. Imputation using the HRC reference panel would not have resolved the issue of missing variants, because certain variants (indels) included in the 1000 genomes project and in the PRS are not available in the HRC panel.

We agree that using a different set of variants could potentially limit the direct comparison with previously published PRS. However, in all our analyses, to enable direct comparison, we also report the associations in European ancestry women by re-estimating the effect sizes of 287-SNP PRSs using the same studies of European ancestry as in the published 313-SNP PRS (results are shown in Table 1 and Table 2). We have highlighted this for the reviewers' attention. In fact, the effect sizes for the 287 and 313 SNP PRS are almost identical. In addition, most of the 26 excluded SNPs are rare in Asians so their contribution, even if they could have been imputed, would be small.

Q4: For the Singapore Chinese Health Study (SCHS), only 229 of the 287 variants that were found to be polymorphic and imputable. Although this is a valuable study because of its prospective nature and information on ancestry, it is a bit confusing to show both the 287-SNP PRS and 229-SNP PRS as main findings. I suggest moving the 229-SNP PRS findings to supplement or make clearer what is the added value of these analyses as main findings. I do not see much value added on tables 1 or 2. Findings on Tables 4 and 5 by ancestry are of interest, are PRS means (SD) on Table 4 based on the 229 or 287 variant PRS?

Response: We agree with the reviewer that SCHS is a valuable study as it provides an opportunity to validate the PRS in a prospective study. Unfortunately the SCHS dataset, which was genotyped with a different array, only provided reliable imputed genotypes for 229 variants. However, the results demonstrate the 229 SNP PRS has very similar discrimination in the SCHS as in the case-control studies showing that the association seen in the case-control studies is replicated in a prospective cohort (albeit that the effect is slightly smaller than the 287 SNP PRS in the case-control studies, as expected). Therefore, we think it is important to retain the 229-SNP PRS results in the main findings. However we have added some explanation

in the footnote of the table to clarify the difference between the 287 and 229 PRS so that the table is self-explanatory.

Table 4 is based on 287-SNPs PRS. We have included footnote in Table 3, 4 and 5 for clarification.

Q5: Are the principal components calculated separately for the 10 Asian studies in BCAC and the three Asian-American studies?

Response: No, the principal components were derived using the full Asian iCOGS and Oncoarray datasets, and then applied to all samples.

Q6: The authors find a smaller RR for family history than previously reported in European-ancestry studies. Are there previous studies in Asian populations that support this difference?

Response: One population-based case control study and one prospective cohort study in Asia estimated the OR/RR for family history to be 1.52 and 2.1 respectively. We have included this information in the discussion (page 14).

Q7: The 287-variant PRS was less predictive of breast cancer risk in the three Asian-American studies than in the studies conducted in Asian (which found no differences between Asians of Chinese, Malay or Indian origin). This is a potentially important finding that requires confirmation in larger studies. Could the authors formally test if these differences are statistically significant in the current population? Based on these findings, can the authors comment on what needs to be done to use PRS in Asian populations in the US? Or other countries?

Response: We had tested for a difference in the PRS effect size based on Asian studies and the effect size from the three Asian-American studies using a Z-test statistic (assuming the logORs are normally distributed) and show that the difference is statistically significant. The result has been highlighted for the reviewer's attention (pages9-10). However, we would like to emphasize that the findings from the three Asian-American studies should be interpreted cautiously. As shown in Figure 1, of the three studies, the largest (LAABC, genotyped using the iCOGs array) showed a significant association with magnitude that is similar to other iCOGs Asian studies. The remaining two studies (one from the US, one from Canada), in which the magnitude of association is smaller compared to other studies, are based on very small sample sizes, and in particular only include a small number of controls. Although we agree that this is a potentially important finding, in the absence of larger north American studies it would be premature to make a definitive statement on the utility of European-ancestry based PRS in predicting risk in Asians in north America.

Minor comments:

Q1: In the imputation for BCAC, the phase of the 1000 genome project should be specified.

Response: We have included the information as suggested (page 16).

Q2: The EM algorithm needs citation

Response: We included the reference as suggested (page 19).

Q3: The version of R and Stata needs to be listed

Response: We have included the version of R and Stata that were used (page 20).

Q4: Indicate population/sample size included in each table/figure or a reference to a table/figure showing the population/sample size.

Response: We have included footnote in each table/figure the reference to Supplementary Table 1 which shows the studies and corresponding sample sizes.

Q5: Table 1- change format to match Table 2 (with less blank cells).

Response: We have updated Table 1 as suggested.

Q6: Consider adding the European estimates to Figure 2

Response: We have included the theoretical estimates of European studies to Figure 2 as suggested by the reviewer.

Q7: Consider adding estimates from American Asians to Figure 3.

Response: The analyses in Figure 3 focus on comparison of the predictive performance of 287-SNP PRSs across the three ethnic groups in Malaysia and Singapore. The corresponding estimates from American Asians are already included in Figure 1.

Q8: Tables 3 and 4- indicate what PRS is shown – 287-SNP PRS?

Response: We have included footnote in Table 3, 4 and 5 for clarification.

REVIEWERS' COMMENTS:

Reviewer #2 (Remarks to the Author):

The authors have addressed my comments adequately

Reviewer #3 (Remarks to the Author):

The authors have made a comprehensive response to the questions raised by the reviewers. The addition of explanatory comments and re-editing has improved the clarity of the paper and there are no significant revisions that I would suggest.

Appreciating that the paper has been well reviewed and revised there were a couple of points that might be addressed for the final text:

Line 297 - the two figures here seem to be for Asians in Asia and Asians in America but it is not actually clear and could be spelled out

Line 414 - it may be worth noting that this figure (20%) would then be around the same as it is in the European population - as written it leaves the impression that this is still exceptional

Line 441 - "the PRS based on large European-ancestry studies may be used as the basis for Asian-specific breast cancer risk prediction models" - although it has been well covered in the text this summary statement probably still needs to have a caveat such as "that take into account the unique distribution of the PRS in the Asian population" to avoid giving the impression that the risk prediction produced by the standard European-trained model (such as generated by BOADICEA or from commercial labs) is 'good enough' for the interpretation of genotype data from an Asian woman.

Reviewer #4 (Remarks to the Author):

The authors have adequately responded to all queries. One point of confusion that remains is what analyses use SNP weights from Europeans vs. SNP weights from Asians and how do these results compare. This needs to be better defined in the methods, figure/table legends and in the results section.

RESPONSE TO THE REVIEW COMMENTS

We appreciate the time and effort taken by the reviewers in reviewing the manuscript.

Reviewer #2

The authors have addressed my comments adequately.

Response: No action required.

Reviewer #3

The authors have made a comprehensive response to the questions raised by the reviewers. The addition of explanatory comments and re-editing has improved the clarity of the paper and there are no significant revisions that I would suggest.

Appreciating that the paper has been well reviewed and revised there were a couple of points that might be addressed for the final text:

(1) Line 297 - the two figures here seem to be for Asians in Asia and Asians in America but it is not actually clear and could be spelled out

Response: We have edited the sentence for clarity as shown below:

“The estimates were similar to those for the 229-SNP PRS in Asian studies (Asian studies from Asia: 1.49 (1.45-1.52); North American studies: 1.33 (1.22-1.45)) but slightly lower than those in the European studies (1.59 (1.55-1.64)).”

(2) Line 414 - it may be worth noting that this figure (20%) would then be around the same as it is in the European population - as written it leaves the impression that this is still exceptional

Response: We have edited the sentence to clarify that the figure (20%) was reported in European studies.

“If the incidence rate reaches that of Western European countries, a similar proportion of women (~20%) would not meet screening threshold at any age.”

(3) Line 441 - "the PRS based on large European-ancestry studies may be used as the basis for Asian-specific breast cancer risk prediction models" - although it has been well covered in the text this summary statement probably still needs to have a caveat such as "that take into account the unique distribution of the PRS in the Asian population" to avoid giving the impression that the risk prediction produced by the standard European-trained model (such as generated by BOADICEA or from commercial labs) is 'good enough' for the interpretation of genotype data from an Asian woman.

Response: We have included in the sentence that PRS developed in European population needs to be calibrated to the Asian population.

“In the meantime, the PRS developed using data from large European-ancestry studies (provided this is re-calibrated to the population being tested) may be used as the basis for Asian-specific breast cancer risk prediction models that include the PRS as well as other predictors of breast cancer risk.”

Reviewer #4

The authors have adequately responded to all queries. One point of confusion that remains is what analyses use SNP weights from Europeans vs. SNP weights from Asians and how do these results compare. This needs to be better defined in the methods, figure/table legends and in the results section.

Response: We have included a paragraph in method to describe how Asian PRSs were derived.

“We compared the predictive performance of the European ancestry-based PRS with PRSs that were previously developed or evaluated in Asian populations. The five Asian population-derived PRSs included 5 SNPs¹⁵, 51 SNPs¹⁷, 44 SNPs¹⁹, 6 SNPs²⁶ and 46 SNPs²⁷. The PRSs were derived using Equation (1) and the corresponding reported in the literature.”